# Addressing Immunization Inequity—What Have the International Community and India Learned over 35 Years?

**DOI:** 10.3390/vaccines11040790

**Published:** 2023-04-04

**Authors:** Lora Shimp, Raj Shankar Ghosh, Katharine Elkes

**Affiliations:** 1JSI Research and Training Institute, Arlington, VA 22202, USA; lora_shimp@jsi.com; 2Public Health Consultant, Delhi 201301, India

**Keywords:** immunization, equity, inequity, India

## Abstract

Countries around the world established immunization programs over 40 years ago to reach all infants. The maturity of these preventive health programs offers some useful learning on the importance of, and components needed for, population-based services to reach all communities. A public health success, ensuring equity in immunization, requires a multi-faceted approach that includes sustained government and partner commitment and human, financial, and program operational resources. Evidence from India’s Universal Immunization Program (UIP) across stabilizing vaccine supply and services, enhancing access, and generating demand for vaccines in the community provides a useful case study. The political leadership in India took advantage of the two decades of learning from polio eradication and focused initiatives, such as the National Health Mission and Intensified Mission Indradhanush, to reach populations with immunization services. With a goal of leaving no one behind, India’s UIP and partners are bringing essential rotavirus and pneumococcal vaccines nationwide, upgrading vaccine cold chain and supply systems with technologies, such as the electronic Vaccine Intelligence Network (eVIN), and optimizing funding for local needs through the Program Implementation Plan (PIP) budgetary processes and building health worker capacities through training, awareness, and e-learning.

## 1. Introduction 

The international community can debate equality and equity definitions and terminology, but fundamentally, are we considerate of individuals and their rights, needs, and convenience to access trustworthy and quality preventive health services, such as immunization? Are our health workers confident, satisfied, and sufficiently and proactively resourced in their work to provide these services to each individual, notably in rural, dense urban, fragile, or emergency settings with limited or no modern technology? These questions are explored in this article, including what it takes to sustain and expand a large-scale public health program, such as immunization, to ensure coverage, quality, equity, and inclusiveness. We reflect on this in the context of India’s Universal Immunization Program (UIP) and the stewardship, policies, and initiatives for vaccine supply and program advances, technology innovations, and the legacy of learnings (such as from the Polio Eradication Initiative) to boost coverage and reduce inequities in routine immunization.

## 2. Background

The global Expanded Program on Immunization (EPI) was established in 1974, with remarkable planning and technology advances with countries, particularly over the last two decades [1]. However, how much of what is on paper or electronic forms or collected via our robust global immunization tracking system, referred to as “WUENIC”, has been proven to consistently reach and incorporate inputs and feedback loops from all users [2]? These users include clients, caregivers, and all cadres of health workers, notably those who are community-based and may be more informally linked with the health system. Immunization programs in countries can incorporate biometrics and machine learning, but are full equity and human interface assured with these technologies to every individual whose data is collected or who uses the data? For sustainability, the backend technology management and access—and the data collected—must be owned, archived, and accessible over many years and as technology advances. This includes access not only for the health system but also by clients, such as parents or caregivers, to show verifiable immunization records for their child’s school entry or by a refugee or immigrant in a stressful transient situation. Are we also addressing the behavioral science and economics of individuals in their decision-making to seek and access immunization services and their ability and comfort to act on those decisions to have themselves and those in their household and communities vaccinated? 

The further one is removed from a problem, the easier it may seem to address. We need to check those assumptions, consider the people-side, and triangulate with qualitative measurements and process indicators, beyond the coverage figures, reporting milestones or quantitative data [3]. This is particularly relevant in the face of weak vaccine-preventable disease surveillance systems that can help to inform immunization program reach in communities. The vast majority of people around the world participate in vaccination services, as shown in WUENIC. However, several decades of polio eradication, the need for repeated measles campaigns in areas with pockets of unvaccinated clusters of people, and the global urgency of the COVID-19 pandemic have shown us that there are no quick fixes. Technology that is not fit-for-purpose nor singular vertical interventions fully meet public health needs of the most vulnerable populations. The global health community and donors need to refocus and sustainably resource preventive health interventions as a collective ‘global good’ for each birth cohort and over the life course, well beyond shortsighted annual funding. Governments and donors also need to reflect collectively on previously agreed upon recommendations, such as those of the Ministerial Conference on Immunization in Africa, to assess and revisit their own commitments [4].

While it helps to have universal terminology, such as the recent use of ‘zero dose’ for infants who have not received their first dose of DPT-containing vaccine, words do not guarantee action, and one year’s success is not indicative of what it takes to maintain and grow a robust system [5,6]. The Immunization Agenda 2030 is ambitious and holistic, but we have learned from the over 35 years of EPIs and previous studies that local operational resources are critical for optimal performance every year [7,8,9]. 

## 3. Success of Routine Immunization in India

Two critical components for addressing equity and moving towards assured immunization program sustainability are the commitment and incorporation of local resources (particularly at subnational levels), and engagement and partnerships with civil society. India provides an interesting case study.

India’s routine immunization program success to date can be summarized around six major milestones:

India’s Universal Immunization Program (UIP) was launched in 1985 by the Indian Government, with prioritized (and annually budgeted and planned) local financial and logistics resources from federal, state, and district levels. This established the system for delivering essential vaccines (such as those preventing diphtheria, pertussis, tetanus, polio, and measles) to infants around the country and tetanus vaccine to pregnant women to prevent neonatal tetanus. Over the years, system strengthening has also increased focus on the following: supply chain to ensure availability of quality vaccines at every level; vaccine safety by augmenting adverse events following immunization (AEFI) surveillance; and data quality and accessibility. Two-way communication between the service provider and the beneficiaries also must be augmented through digital web-based platforms like the Mother Child Tracking System and availability and use of Maternal and Child Health cards that include all antigens and reminder dates. Additionally, particularly in the last 10 years, UIP has collaborated with Gavi, the Vaccine Alliance, and partners to augment skills of health workers and front-line program managers, such as via initiatives like Routine Immunization Skills Enhancement [10].

In 1995, India launched the nationwide pulse polio immunization program, including National Immunization Days for supplemental polio vaccination. These efforts, linked with routine vaccination that also emphasized birth dose polio vaccination, encouraged multi-stakeholder coordination, program innovation, and community mobilization and engagement at every level of program planning and implementation. This included important collaboration with civil society partners, such as Rotary and the multi-partner Social Mobilization Network led by UNICEF and the Core Group Polio Partners [11]. India’s recognition of being polio free in 2014 also elevated the value of vaccination and contributed to a shift in focus on routine immunization [12]. 

To further address equity in reaching often missed or underserved communities, India launched the National Health Mission in 2013 [13]. The National Health Mission was a bold step towards integration of immunization with other program deliverables in Primary Health Care. The program also integrated two previously vertical and siloed initiatives that began in 2006: National Rural Health Mission and National Urban Health Mission. 

The Mission Indradhanush (MI) Program, launched in 2014, and the subsequent launch of Intensified Mission Indradhanush (IMI), launched in 2017, were designed to address vaccine inequity in a subset of districts and facility clusters across geography and gender, based on evidence from data collected from districts. Important within both initiatives is the role of civil society as key partners, including engaging the accredited social health activist (ASHA) program for linking missed communities with immunization services [14]. These initiatives have also contributed to surveyed fully immunized coverage (FIC) increases, as shown in Figure 1 and Table 1, with national FIC coverage at 76.4% for 12–23-month-olds from the most recent NFHS-5, 2019–2021 [15].

In recent years, the UIP has expanded to include rotavirus vaccine, pneumococcal conjugate vaccine, inactivated polio vaccine, measles-rubella vaccine, and the Japanese Encephalitis vaccine (for adults). Political and bureaucratic administrator interest has been high at all levels, including from the Prime Minister. UIP and several donor and resource partners supported these introductions at a national scale through sophisticated epidemiology, investment evidence and technical support, and civil society partner engagement for communications and confidence and trust building. For example, donors, such as the Bill and Melinda Gates Foundation, Gavi, and the Vaccine Alliance, provided complementary support for the vaccine rollouts, including additional technical assistance via the Immunization Technical Support Unit and partners, such as UNDP, UNICEF, WHO, John Snow Inc/India, Clinton Health Access Initiative, and others. Additionally, the program was emboldened by domestic manufacturing of the vaccines and a committed supply for scale up. The new vaccine rollouts also provided opportunity for strengthening health systems with technologies, such as the electronic Vaccine Intelligence Network (eVIN), Vaccine Safety Monitoring, and the National Cold Chain Management Information System (NCCMIS) for real time cold chain monitoring and management decisions. 

COVID-19 vaccination necessitated rapid, wide-scale digital technology to facilitate vaccine access across the majority of India’s population. The COVID-19 vaccination tracking software, known as CoWIN, enabled citizens to choose their vaccination place and time at their convenience with strong community acceptance as evidenced in the high COVID-19 vaccination rates in India [16]. Expansion of the tool with Indian resources is anticipated to benefit routine immunization equity and coverage, enabling health workers and citizens to track routine immunization through the digital application known as UWIN.

As the COVID-19 pandemic response shifts, India is reviving and sustaining its routine immunization coverage improvement program with experience, equity, evidence, and empowerment.

India has gained extensive experience from large scale polio and measles vaccination campaigns and conducting the world’s largest routine immunization program (Universal Immunization Program) to reach approximately 26 million infants and 30 million pregnant women. As noted, Intensified Mission Indradhanush and COVID-19 vaccination have also contributed to equity in previously under-served communities and populations. The experiences gained from these programs cut across vaccine supply, access, and community mobilization and have been institutionalized in India’s immunization and health systems. A few examples of this institutionalization include vaccination microplans that are part of annual health performance implementation planning, alternate vaccine delivery systems, house-to-house immunization campaigns, and community radio for peer-to-peer conversation in the community. 

To address the equity gap, India is tailoring efforts across geographies, gender, and socio-economic strata. During COVID-19 waves in India, some of the most heavily disrupted populations were remote, economically challenged people with specific needs (such as the differently-abled), the transgender community, and populations that migrated from their workplaces. These populations also deserve attention as primary health care programs, such as immunization, adjust in the post COVID-19 phase. States are partnering with community-based organizations and civil societies to tailor services that will enable better access to populations with specific and special needs. Examples of such initiatives are the iHEAR project of Sangath that generated evidence around challenges faced by the disabled and transgender communities during COVID-19 vaccination and the Vaccine on Wheels project of Jivika that provided doctor-supervised mobile medical units for immunization. 

To inform and mobilize evidence-based action by generating high-quality digital data, India is strengthening its laboratory-supported Vaccine Preventable Disease (VPD) surveillance with the help of domestic institutions, such as the National Centre for Communicable Diseases and the National Public Health Support Program of the World Health Organization, India Country Office. The data generated from an empowered VPD Surveillance will, in turn, help health workers to take strategic, timely action to address coverage inequity and improve the quality of immunization services. 

A key lesson from COVID-19 vaccination was the ability to integrate the use of digital technology, such as via CoWIN and its mobile app, for empowering clients with vaccination information and records. CoWIN generated high-quality real-time data for communities to make informed choices about where and when to organize and receive services; for health workers to track, record, and report vaccination; and for health authorities to take timely action. With lessons from the wide acceptance of CoWIN by the community for COVID-19 vaccination, India is planning to empower its communities with digital UWIN technology. The technology will empower service providers with digitalization of immunization records for tracking, recording, and reporting immunization coverage. For the community, the application will provide flexibility by allowing them to choose the location where they want to receive vaccination services and to have a digital record of their data and vaccination history that can be downloaded and saved. The digital record of citizens will be directly linked to their Ayushman Bharat Health Account (ABHA). The ABHA account will uniquely identify every registered individual as a participant in India’s digital healthcare system. 

## 4. Discussion

Despite the remarkable achievements in India’s vaccination efforts over the last several years, equity challenges remain to consistently and sustainably reach the most underserved populations. This is not unique to India. Shifts in immunization program focus and strategy, as outlined in the Immunization Agenda 2030, need to align with what immunization data are showing us on continuing inequities. These shifts include addressing disparities across countries and within large countries, such as India, through tailored approaches that are designed with and resourced to the specific fragile and conflict-affected, rural (remote and non-remote), and urban populations. 

How can public health programs and donors adapt learning from the Indian immunization evolution and apply the latest equity approaches and tools, particularly in lower-resource settings? A key priority is to foster coordination and long-term resourcing with local institutions that are best placed to generate workable solutions with their populations. This can be achieved through supporting and partnering with civil society networks, particularly those that are established and have a track record of managing resources. As noted earlier with Rotary’s involvement in polio eradication, civil society networks are more likely to garner local support, including for day-to-day operational funding, if they are part of planning and monitoring. This includes having access to data and opportunities for regular review meetings with health service representatives. The broader public and private sector health practitioner networks, such as the International Pediatric Association and the International Council of Nurses, also play a critical role in linking people with services for a positive experience of care.

Why is this important? As the COVID pandemic demonstrated, health workers are not only essential for preventing and managing outbreaks but they are also clients themselves. However, oftentimes health systems are not meeting their basic needs for a positive service experience, such as balanced workloads and sufficient supplies. In their delivery of immunization and primary health care, health worker networks will benefit from further adaptation of existing resources that have shown potential across many countries, such as the Reaching Every District and Tailoring Immunization Programmes guidance [17,18]. Pre-service and in-service training and on-the-job learning and mentoring can integrate the fundamentals of immunization service planning and service experience [19,20]. Figure 2 provides a visual example of service experience components that consider the needs of both health worker and service recipient clients. 

As mentioned previously, funding for operational resources is also critical and requires a paradigm shift back to the fundamental platform of a functional public health program. Donors should require—and hold themselves accountable—in building health systems, such as USAID’s commitment to championing global health and the health workforce in their 2024 budget allocation. This includes funding and monitoring innovations that embody frameworks for local ownership and equity analyses that involve sufficiently representative populations that lack technology access. Indicators, such as consistent availability of data minutes and evidence of use of a mobile device, should be required, not just ownership of a mobile device. Equitable sustainability also requires partnering with local institutions and engaging with communities, which often takes more time and investment but is arguably more likely for public health programs to be able to maintain, particularly in lower income countries.

## 5. Conclusions

The various technical, operational, and resourcing approaches noted in this article take time to implement and to demonstrate impact on reducing inequity, as also shown in the evolution of India’s immunization program. Nonetheless, important learnings can be adapted now for incrementally improving immunization services, quality, and access with populations. As annual coverage data provide a time-limited snapshot, immunization programs and donors will benefit from triangulating coverage data with process indicators and trend analyses. In addition, sustained immunization program success requires continuing political and administrative buy in, technical quality, program review at the district level upwards, and community partnerships. As the Immunization Agenda 2030 progresses, the global immunization community and countries can benefit by tailoring their immunization equity strategies from previous experiences, such as the components shown in the India example, and incorporating approaches that include behavioral science and person-centered care to support and empower health workers and clients.

## Figures and Tables

**Figure 1 vaccines-11-00790-f001:**
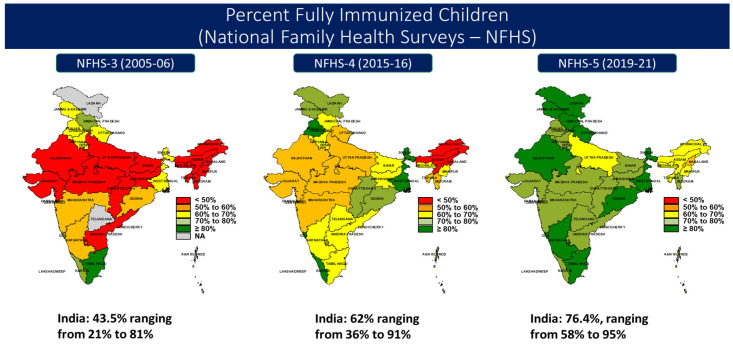
Percent of Fully Immunized Children (National Family Health Surveys—NFHS).

**Figure 2 vaccines-11-00790-f002:**
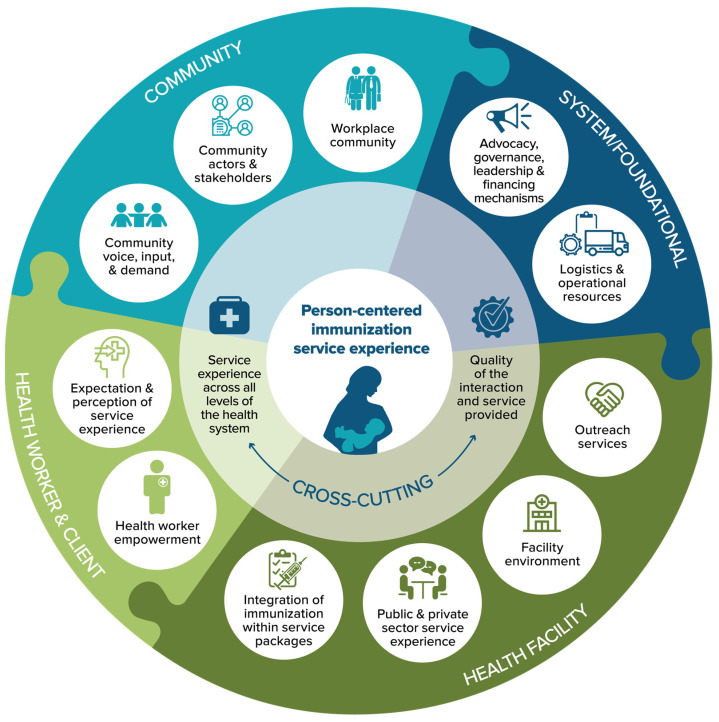
Person-centered immunization service experience graphic.

**Table 1 vaccines-11-00790-t001:** India fully immunized coverage (12–23-month-olds, card and maternal recall) by state from National Family Health Surveys.

Fully Immunized Coverage	NFHS-3 (2005–2006)	NFHS-4 (2015–2016)	NFHS-5 (2019–2021)
Average	43.5%	62%	76.4%
Range (by state)	21–81%	36–91%	58–95%

## Data Availability

No new data were created or analyzed in this study. Data sharing is not applicable to this article.

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
