# Peer review of "Addressing Immunization Inequity—What Have the International Community and India Learned over 35 Years?"

_vaccines, 2023, doi:10.3390/vaccines11040790_

Round 1

Reviewer 1 Report

The article brings interesting information regarding India's vaccination programs and progresses achieved in past years.

It is well written and based on evidence and on knowledge of the writers.

However, in my view, it is quite descriptive and has no input from contradictory views or shows challenges or problems regarding efforts to achieve equity in vaccination for the whole country.

The paper could benefit and be enriched by scientific literature that discusses challenges of universal coverage in low resources countries.

Author Response

The authors thank the reviewer for these suggestions. As seen in the attached version with track changes, we have added the following to the article: - detail on addressing inequities in India added to the Abstract section - expansion of India context in the Introduction - references on donor accountability and community engagement for sustainability included in paragraph before Conclusion section (lines 240-241 and 246 in the updated version)

Reviewer 2 Report

This very well-written opinion article assesses the status and achievements of India's Universal Immunization Program.  A strong history is presented, along with the current challenges in addressing preventable disease surveillance.  Here is a list of suggestions for very minor refinement:

(1) Abstract: Consider elaborating on how inequities can be addressed in India (e.g., National Health Mission, MI Program, empowerment, etc.); also, consider indicating in the abstract that the article addresses health workers (Lines 8-14)

(2) Introduction - This section is well-written (lines 18-23).  Consider expanding with 1-3 additional sentences with context specific to India 

(3) Background -Very interesting commentary on the status of digital technology and VPDS surveillance. 

(4) Success -   Great infographics.  Would be good to see an example of donor accountability (line 216)  and/or a suggestion for equitable sustainability (line 221) 

Author Response

The authors agree that challenges in achieving immunization equity should be added. To address this, we have added a paragraph and reference on recent equity analysis. This is now the opening paragraph in the "4. Discussion" section, lines 202-210 in the updated version.  
